# Unraveling the Protective Effects of Cognitive Reserve on Cognition and Brain: A Cross-Sectional Study

**DOI:** 10.3390/ijerph191912228

**Published:** 2022-09-27

**Authors:** Dorota Szcześniak, Marta Lenart-Bugla, Błażej Misiak, Anna Zimny, Marek Sąsiadek, Katarzyna Połtyn-Zaradna, Katarzyna Zatońska, Tomasz Zatoński, Andrzej Szuba, Eric E. Smith, Salim Yusuf, Joanna Rymaszewska

**Affiliations:** 1Department of Psychiatry, Wroclaw Medical University, Pasteura 10, 50-367 Wroclaw, Poland; 2Department of General and Interventional Radiology and Neuroradiology, Wroclaw Medical University, Borowska 213, 50-556 Wroclaw, Poland; 3Department of Social Medicine, Wroclaw Medical University, Bujwida 44, 50-345 Wroclaw, Poland; 4Department and Clinic of Otolaryngology, Head and Neck Surgery, Wroclaw Medical University, Borowska 213, 50-556 Wroclaw, Poland; 5Department of Angiology, Wroclaw Medical University, Borowska 213, 50-556 Wroclaw, Poland; 6Department of Clinical Sciences and Radiology, Hotchkiss Brain Institute, University of Calgary, Calgary, AB T2N 4N1, Canada; 7Population Health Research Institute and Department of Medicine, Faculty of Health Sciences, McMaster University, Hamilton, ON L8L 2X2, Canada

**Keywords:** white matter hyperintensities, brain reserve, cognitive functions

## Abstract

This study aimed to investigate the hypothesis that brain maintenance expressed in white matter hyperintensities and brain reserves, defined as gray and white matter volumes, mediate the association between cognitive reserve (CR) and cognitive performance. A cross-sectional population-based observational study was conducted, and the final study sample consisted of 763 participants (282 men and 481 women) with a mean age of 61.11 years (±9.0). Data from different categories were collected from study participants, such as demographic, lifestyle, medical, and psycho-social characteristics. All participants underwent a detailed psychometric evaluation (MoCA and DSST) followed by a brain MRI. Volumetric measurements of the total gray matter (GMvol), total white matter (WMvol), and white matter hyperintensities (WMHvol) were performed using the Computational Anatomy Toolbox 12 (CAT12) and Statistical Parametric Maps 12 (SPM12) based on 3D T1-weighted sequence. Significant direct and indirect effects of cognitive reserve on cognitive functioning were measured with both scales—the MoCA and DSST. In each mediation model, the volumes of WMH and GM were significant mediators for the association between cognitive reserve and cognitive performance. This study confirms the importance of strengthening the cognitive reserve in the course of life through potentially modifiable effects on both cognition and the brain.

## 1. Introduction

The topic of individual differences in cognitive functioning and brain changes in the aging process is broadly discussed in the literature and research [1]. Since dementia research has focused more on preventive and protective factors of cognitive impairments in recent years, the cognitive reserve (CR) has become an essential phenomenon under investigation as it shows important associations with the brain reserve (BR) and brain maintenance (BM). It emerged from the observation that adverse changes are visible in the brain’s structure and consistently reflected in cognitive decline [2,3]. Although cognitive reserve remains an abstract concept, its various definitions and operationalizations have been developed by previous studies [4]. The cognitive reserve can be conceived as a phenomenon or as a mechanism. Recognized as a mechanism for coping with brain damage or cognitive impairments, cognitive reserve coincides with the concept of brain reserve, which according to Stern et al. (2020) [3] refers to neuroanatomic brain structures. In turn, brain reserve is perceived as a surplus of competence for the maintenance of cognitive functioning. Both cognitive reserve and brain reserve increase individual resistance to the aging process and the appearance of clinical symptoms [5]. While brain reserve is easier to operationalize as it relates to measurable structural features of the brain (white and grey matter volume, and white matter integrity), cognitive reserve remains a more difficult concept to grasp and measure [4], especially, since one more closely related phenomenon is recognized, namely brain maintenance. It refers to the reduced development of age- or disease-related brain changes and reduced pathology accumulation over time, which for example, can be reflected in vivo by fewer white matter hyperintensities (WMH) in magnetic resonance imaging (MRI) [6].

These observations imply a vague character of the cognitive reserve construct and a lack of consensus on its crucial components [7]. Therefore, this concept pinpoints the importance of numerous factors influencing the level of cognitive reserve, which are commonly known as its proxies or markers. The most frequently studied factors recognized as markers for cognitive reserve include intelligence, education, and occupation [2,8]. Recently, various lifestyle factors have also been considered, examining their impact on the cognition and brain, such as physical activity [9,10], cognitive activity, social engagement, or leisure activities [11,12,13]. Recent studies have focused more on social factors influencing cognitive reserve, mainly their positive impact on cognition, such as social engagement, social activities, supportive personal network with various types of relationships, a larger number of significant family members, and a larger number of close friends [2,12,13,14,15,16]. This is in line with the observed cognitive decline in those with few family ties, fewer friends, and less social support [12]. However, until now, the central attempts to estimate cognitive reserve on cognition have often been limited to a single marker analysis.

According to Pettigrew and Soldan (2019) [4], the complex nature of the structure of cognitive reserve and the still unknown mechanisms of its relationship with the brain and cognition require further exploration of the protective role of its potential indicators. Previous studies have separately described the relationship between cognitive reserve and brain reserve markers measured by white and grey matter volumes [17,18,19,20]. However, studies investigating brain maintenance have provided mixed findings [21,22,23,24].

Hence, the present study aimed to investigate possible mechanisms underlying the association between selected measures of cognitive reserve and cognitive performance. Specifically, we tested the hypothesis that brain maintenance (in terms of WMH) and brain reserve (defined as gray and white matter volumes) mediate the association between cognitive reserve and cognitive performance.

## 2. Materials and Methods

### 2.1. Population

The study population was derived from the Polish sample of the multinational PURE cohort study (Prospective Urban Rural Epidemiology Study) that recruited 1269 inhabitants of Lower Silesia (Wroclaw and the surrounding villages). According to the global PURE study protocol, these participants were approached every three years until the 9th follow-up examination. In the last wave of the study conducted between 2016 and 2019, the cognitive functioning assessment protocol and MRI examination were included. The cross-sectional analysis of participants enrolled in the 9th follow-up considered the assessment of cognitive reserve markers about brain reserve, brain maintenance (MRI data), and cognition.

Of the 1269 baseline participants, 327 did not meet inclusion criteria or refused to participate in the 9th follow-up; 7 did not complete cognitive assessment; 66 were unable to complete MRI; and 57 were excluded due to missing brain volumetric data or brain pathologies other than WMH, with 4 excluded due to motion artifacts on the MR images, resulting in 763 participants (282 men and 481 women) (Figure 1).

### 2.2. Cognitive Reserve and Cognition

Data from different categories were collected from study participants at enrolment and during the follow-up: demographic, lifestyle, medical, and psycho-social characteristics. According to earlier scientific reports, the following analysis includes data that are important markers of CR. As in the study by Durrani et al., 2021, a slightly modified method of determining the CR score was used in this study [25]. The CR score (range 0–8 points) included (1) participation in social groups, (2) being married or in a partnership, (3) engagement in moderate or vigorous physical activity in leisure time, (4) being employed, (5) graduation from high school or trade school (2 points), and (6) graduation from college or university (2 points), as supported by recent studies [9,10,11,12,13,26].

Qualified psychologists conducted the assessment of cognitive functioning with standardized psychometric tools: The Montreal Cognitive Assessment (MoCA) [27], Wechsler Adult Intelligence Scale 4th Edition (WAIS) [28], and the Digit Symbol Substitution Test (DSST) [29]. The MoCA is a screening tool for a diagnosis of mild cognitive impairment (MCI) and is known to reflect CR more sensitively than the Mini-Mental State Examination (MMSE) [30]. The cut-off is set at <26 points; in Polish adaptation, a 1-point correction was used for people with education below 12 years. The DSST is a practical and sensitive neuropsychological test to detect cognitive impairments across three cognitive domains: processing speed, executive functioning, and working memory. The score is the number of symbols matched to the digits in a limited time of 2 min.

### 2.3. Brain Reserve and Brain Maintenance

All brain MR examinations were performed using the same 1.5 Tesla MR scanner (GE, Signa Hdx). The MRI sequences included axial dual echo T2/PD weighted images (TR = 2720 ms, TE = 88/8.8 ms, ET (Echo train length) = 12, FOV = 240 × 240 mm, slice thickness = 3.5 mm, matrix = 256 × 256, NEX = 1), axial fluid attenuated inversion recovery sequences (FLAIR) (TR = 8.800 ms, TE = 145 ms, TI = 2.200 ms, FOV = 240 × 240 mm, slice thickness = 3.5 mm, matrix = 256 × 256, NEX = 1), diffusion weighted imaging (DWI) (SE/EPI, TR = 10,000 ms, TE = 107, FOV = 240 × 240 mm, slice thickness = 3.5 mm, matrix 256 × 256), susceptibility weighted imaging (SWI) 3D (TR = 73.9 ms, TE = 47.4 ms, FOV = 240 × 168 mm, slice thickness = 3.5 mm, matrix 256 × 256, NEX = 0.7, flip angle = 20), and high resolution T1-weighted images (FSPGR, TR = 8.3 ms, TE = 3.2 ms, TI = 650 ms, FOV 240 × 240 mm, slice thickness = 1.0 mm, matrix 256 × 256 mm, NEX = 1, flip angle = 12).

Visual evaluation of MR scans: all MR scans were evaluated by experienced neuroradiologists, and patients with brain lesions other than WMH were excluded from the study. White matter hyperintensities were graded on the FLAIR images using the Fazekas scale of 0–3 [31,32], separately for periventricular and subcortical locations. The combined Fazekas score (periventricular + subcortical) was calculated (0–6 points). Two independent radiologists rated all MR scans using the same criteria for WMH rating. The inter-rater reliability was established with a kappa coefficient of 0.88. Despite the good inter-rater reliability, all scans that showed any discrepancy between the two raters were re-evaluated, and the final score was established by consensus.

Volumetric measurements: data processing workflow consisted of voxel-based processing using the Computational Anatomy Toolbox 12 (CAT12, Structural Brain Imaging Group, University of Jena, Jena, Germany) and the Statistical Parametric Maps 12 (SPM12) software (version 12, The Wellcome Centre for Human Neuroimaging, UCL Queen Square Institute of Neurology, London, UK). Detailed methodology of volumetric postprocessing has been published elsewhere [33]. Final volumetric measurements consisted of the volumes of the total gray matter (GM volume), total white matter (WM volume) as the representation of brain reserve, and white matter hyperintensities (WMH volume) as the operationalization of brain maintenance. They were estimated as direct volumes in mm^3^.

### 2.4. Ethical Approval

The study protocol received approval from the local Bioethical Committee (approval no.: KB-32/2016). All participants in the study completed a written informed consent form as stipulated in the ethical approval.

### 2.5. Statistical Analysis

Descriptive statistics of the demographic data, questionnaires, and brain volumetry results were presented as mean, standard deviation, or several cases with percentages. An analysis of between-group differences in continuous variables and brain volumetry between age groups were performed using the Kruskal–Wallis test with post hoc analysis using the Holm correction.

A mediation analysis was performed to analyze the effect of cognitive reserve on MoCA and DSST with grey matter volume, white matter volume, and WMH volume as mediators. It was performed using the structural equation modeling framework of the Lavaan package [34]. Estimation of indirect effects was performed using non-parametric bootstrap with a sample size of 1000. Every model was adjusted for age, sex, and education (higher-level vs. other than higher). Figure 2 shows the simplified mediation model analyzed in this study. Different models were performed for mediation through GM and WM. All calculations were performed using the R package for Windows (version 4.1.2, Statistical Computing, Vienna, Austria) [35]. Results were considered significant when the *p*-value was <0.05.

## 3. Results

### Study Sample

The general characteristics of the study sample are presented in Table 1. The majority of participants were women, were people in a marriage or partnership, were employed, had secondary or higher education, were not engaged in moderate physical activity in leisure time, and were not participating in regular social activities. The mean age of the participants was 61.11 years (±9.0). The mean general cognitive functioning score measured with the MoCA test was 25.91 (±2.7), where 39% (n = 296) of the study participants obtained scores indicating the presence of cognitive impairment (MoCA < 26). The mean DSST score was 60.18 (±14.9). The MRI assessment showed that the mean WMH volume was 1.95 mm^3^ (±2.4), and the white matter and grey matter volumes were 501.07 mm^3^ (±61.3) and 572.72 mm^3^ (±54.4), respectively.

The analysis of decades of age revealed that age was significantly associated with cognitive functioning (the MoCA and DSST scores), and the WMH volume and brain reserve were measured as grey matter and white matter volumes (Table 2). Moreover, the analyses of differences between the selected age groups indicated that cognitive performance and brain reserve decreased significantly with each decade while the WMH volume increased.

The results of mediation analysis are presented in Table 3. Significant direct and indirect effects of cognitive reserve on cognitive functioning were measured with both scales—the MoCA and DSST. In each model, WMH and grey matter volumes were significant mediators for the association between cognitive reserve and cognitive performance. No significant effects of white matter volume as a mediator were found.

## 4. Discussion

In this study, we introduced a complex measure of cognitive reserve consisting of selected single markers, described and validated by recent studies [9,10,11,12,13,26]. This perspective was already implemented and described by Durrani (2021) [25]. However, in the context of understanding the cognitive reserve, it serves as an innovative approach. Previous measures for cognitive reserve have often been limited to single-item markers and newer approaches have severe problems such as residualization [36].

Considering the above perspective and the interaction approach tested with the mediation model, this study suggests that cognitive reserve exerts a protective role on cognition by affecting brain reserve and brain maintenance. Grey matter volume is a significant marker of brain reserve, commonly conceived as a neurobiological resource of the individual, confirmed in the present study [3]. That is in line with a large body of research indicating its key role in aging processes and decline in cognitive functioning [17,18,19,20]. Thus, our results support previous findings suggesting that cognitive functioning is associated with grey matter volume among the adult population. Importantly, we showed that a higher cognitive reserve score directly affects greater grey matter volume, which is associated with better cognitive functioning. This finding is consistent with findings from the study by Conti et al. (2021) [37] showing that higher cognitive reserve is related not only to structural but also to functional markers of the brain, which can account for the level of cognitive functioning. Moreover, the authors suggested that higher cognitive reserve promotes brain plasticity and optimization of brain functioning. Similar conclusions were drawn by Kwak et al. (2020) [21], who highlighted the cognitive reserve effect on the relationship between grey matter volume and verbal episodic memory that was further moderated by age in elderly individuals. However, the studies described above have mainly considered the brain reserve markers without the simultaneous analysis of the brain maintenance included in one model. Additionally, the knowledge resulting from previous research remains fragmented. Thus, the interaction approach is the main added value of the present paper.

In our study, we analyzed whether the effect of cognitive reserve on cognition was also mediated by brain maintenance, recognized as the lower level of pathology accumulation (measured in this case by WMH volume). The WMH and grey matter volumes were significant mediators for the association between cognitive reserve and cognitive functioning in each model, as measured by the MoCA and DSST. This means that modifiable markers (considered here as indicators of cognitive reserve), i.e., education, occupation, physical activity, social participation, and marital status, on the one hand, are positively related to individual brain reserve and, on the other hand, can serve as protective mechanisms against neuropathology (brain maintenance), for which age is one of the greatest risk factors. While much data show the relationship between cognitive reserve, brain reserve, and cognition, no consistent results support the relationship between cognitive reserve and brain burden of pathology. In Christensen’s study (2007) [22], no association between cognitive decline and the WMH volume was found. Interestingly, the authors suggested that neither education nor any other cognitive reserve measures, including intelligence, were protective factors for cognitive impairments in individuals with high levels of brain atrophy. Other researchers demonstrated that higher WMH volume is associated with poorer cognition, while greater [25] cognitive and brain reserve are related to better cognition [23]. Nevertheless, the authors did not analyze the relationship between cognitive reserve and the WMH volume. The review conducted by Pinter et al. (2015) [24] based on six studies confirmed that higher cognitive reserve (measured with the level of education) attenuates the negative impact of the WMH on cognition. Furthermore, a recent analysis made by Durrani (2021) [25] showed that vascular brain injury and cognitive reserve markers are associated with cognition. However, according to the authors, the effects were independent, and no relationship was found between cognitive reserve and covert vascular brain injury. The difference between their findings and those obtained by the present study may result from different operationalization methods of the brain burden. Durrani et al. (2021) [25] found that the cognitive reserve and WMH had independent effects on cognition, without evidence of moderation, but did not test whether WMH mediates the effect of cognitive reserve on cognition and uses a visual rating scale for WMH rather than quantitative volumes, which might have better sensitivity in the analyzes. However, it should be noted that its relation remains weak despite the significant relationship between cognitive reserve, WMH volume, and cognition. Moreover, the results of the mediation model showed both direct and indirect effects of cognitive reserve on cognitive functions. This implies an important conclusion on other processes potentially mediating the relationship between cognitive reserve and cognition, not included in the present analysis. It is crucial to consider the cognitive reserve as both a phenomenon and a mechanism to build hypotheses on these processes, while in our study, we applied the cognitive reserve operationalization based on the phenomenon perspective, the results from the mediation model also indirectly support the mechanistic conceptualization of the CR. In the mediation analysis, we obtained results showing partial mediation. Thus, the results seem to point out that there is a unique component of cognitive reserve that remains to be described (i.e., neural compensation). Besides the obvious fact that there are also other measures of brain reserve and brain maintenance that were not analyzed in this study. Thus, in the present results, the cognitive reserve mechanism is visible in the observation that brain reserve and brain maintenance are not complete mediators for cognitive functions. The cognitive reserve mechanism refers to the individual ability to achieve optimal neuronal performance during cognitive processes due to using various cognitive strategies [2]. In the case of natural aging, the cognitive reserve acts as a neural reserve, reflecting the brain’s plasticity, which activates pre-existing cognitive networks. In contrast, in the context of cognitive impairment present in neurodegenerative diseases, cognitive reserve works as neural compensation, recruiting the needed networks to neutralize the effects of neuropathology [38]. An example of such a mechanism is a person who maintains function or does not show any clinical expression despite dementia-related pathology [14]. Hence, two people with the same brain capacity may differ in the effectiveness of using brain structures. In this light, cognitive reserve, seen as a mechanism, helps explain research results showing individual differences with the same type of brain damage and brain reserve, indicating various neural processing and compensation strategies. Factors such as higher synaptic density and better functioning of brain networks are hypothesized to underlie the protective processes related to cognitive performance and the risk of dementia [24].

In addition, the psychophysical state cannot be ignored in the context of both using the cognitive reserve and the level of cognitive functioning. The presence of depressive symptoms may modify the cognitive reserve potential and interfere with cognitive functioning [39,40]. However, because the study sample was relatively homogeneous in terms of the presence of depressive symptoms, and the level of these symptoms was low, we can consider this confounding factor non-significant in this analysis.

The strength of our study is the holistic approach taken to unravel the mechanism crucial for the phenomenon of the cognitive reserve. On the one hand, the individual level of cognitive reserve is seen as an essential factor that can influence brain reserve. On the other hand, it is seen as playing a role in protective mechanisms against neuropathological changes that develop with age. We also recognize the limitations of this study. First, the cross-sectional design limits the possibility of formulating causal associations. We assume a specific directivity of cognitive reserve effect on cognition based on the proposed models. These assumptions have their theoretical foundations and result from the previously conducted research. However, the potentially reversed impact of the relationship between cognitive reserve, brain reserve, brain maintenance, and cognitive functions cannot be ignored. The cause-and-effect process can only be traced by analyzing the trajectory of changes observed in longitudinal studies. The PURE was initially designed as a cohort study for cardiovascular disease, and cognitive assessment and MRI data have only been collected once, so we could not use the results of the longitudinal analyzes. This limitation highlights a significant need for further research, which should focus on analyzing the trajectories of changes in terms of all the variables indicated in our model. The second limitation of our study is the composition of the cognitive reserve score. The cognitive reserve markers were selected from the variables available in the PURE cohort study and, therefore, did not reflect an ideal CR measure. They are based on self-assessment survey data, and it should be noted that aspects such as physical activity or social participation should ideally be based on more objective methods. Using a more rigorous methodology and standardized tools to measure selected cognitive reserve markers to obtain in-depth knowledge would be crucial.

Moreover, we decided to include social participation as a marker of cognitive reserve, being aware that this might seem like a reductionist approach. Current clinical knowledge and research indicate the need to study aspects of social health as a separate phenomenon affecting cognitive reserve and brain reserve. Still, the limited data available did not allow this approach to be applied. Moreover, another limitation of the study is the lack of representativeness of the study population, which significantly impacts the possibility of generalizing the obtained results to the general population.

## 5. Conclusions

This study confirms the importance of strengthening the cognitive reserve across the lifespan through a potentially modifiable effect on both cognition and the brain. Targeting potentially modifiable lifestyle factors, such as education, work activity, physical activities in leisure time, and social health aspects could exert a beneficial impact on cognitive health in later life and delay the onset of dementia. However, it is important to note the importance of introducing these activities in a way of life from early adulthood through middle age and old age as changes in the brain related to cognitive reserve occur much earlier in life and, thus, can be used as a neural compensation when needed. Moreover, the analysis of individual cognitive reserve markers gives a clear direction to developing guidelines for preventive strategies and the construction of early psychosocial intervention in dementia, which can be successfully developed both in clinical practice and further research.

## Figures and Tables

**Figure 1 ijerph-19-12228-f001:**
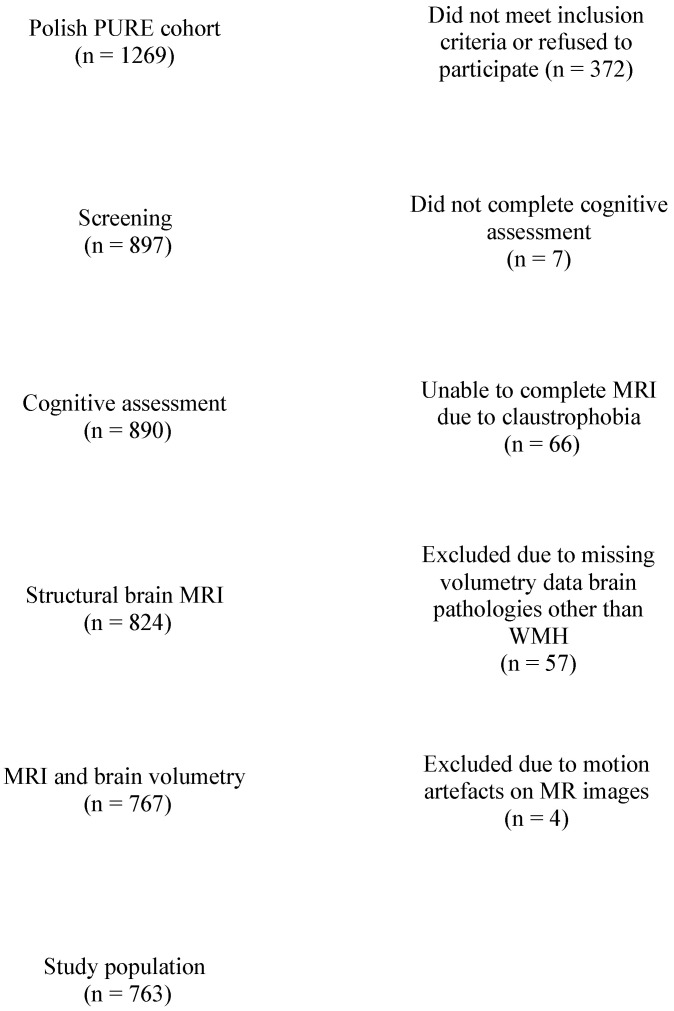
Flow diagram of the study sample.

**Figure 2 ijerph-19-12228-f002:**
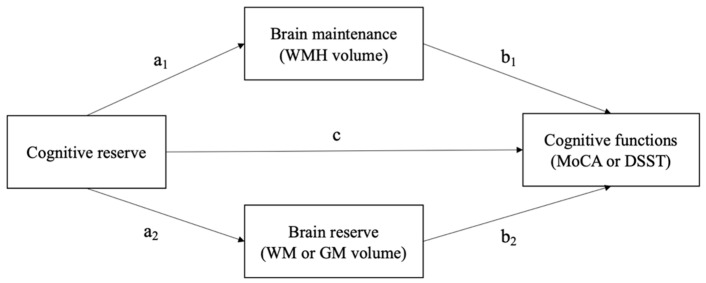
A simplified mediation model tested in this study. Abbreviations: WMH = white matter hyperintensities; GM = grey matter; WM = white matter; MoCA = Montreal Cognitive Assessment; DSST = Digit Symbol Substitution Test.

**Table 1 ijerph-19-12228-t001:** Characteristics of the study population (N = 763).

Characteristics		No. (%) or Mean (SD)
**Age**	Mean in years	61.11 ± 9.0
**Sex**	Female	481 (63)
	Male	282 (37)
**Living location**	Urban	598 (78)
	Rural	165 (22)
**Cognitive reserve (CR)**		
**Education level**	Primary	38 (5)
	Trade school	78 (10)
	Secondary/High school	333 (44)
	College/University	314 (41)
**Marital status**	Common married/living with the partner	578 (75.5)
	Divorced	68 (9)
	Never married	60 (8)
	Widowed	56 (7)
	Unknown	1 (0.5)
**Employment**	Yes	523 (79)
	No	240 (30)
	No date	3 (1)
**Leisure time physically active**	Yes	73 (14)
	No	459 (60)
	Unknown	231 (26)
**Participation in social groups**	Yes	95 (13)
	No	668 (87)
**Cognitive functioning**	MoCA < 26	296 (39)
	MoCA mean score	25.91 ± 2.7
	DSST mean score	60.18 ± 14.9
**Brain maintenance (WMH)**	Fazekas grade	
	(periventricular + subcortical)	
	0	169 (22)
	1	173 (23)
	2	310 (41)
	3	55 (7)
	4	36 (4.5)
	5	8 (1)
	6	12 (1.5)
	WMH volume	1.95 ± 2.4
**Brain reserve**	WM volume	501.07 ± 61.3
	GM volume	572.72 ± 54.4
**Depressive symptoms**	CES-d mean	10.0 ± 8.1
**General activity**	SAGE mean score	2.47 ± 2.9

Abbreviations: MoCA = Montreal Cognitive Assessment; DSST = Digit Symbol Substitution Test; WMH = white matter hyperintensities; BMI = Body Mass Index; CES-d = The Center for Epidemiologic Studies Depression Scale; SAGE = The Standard Assessment of Global Activities in the Elderly. Continuous values are mean (SD). Brain volumes are expressed in [mm^3^].

**Table 2 ijerph-19-12228-t002:** Cognitive functioning, brain maintenance, and brain reserve by age decades.

Age Group	39–49	50–59	60–69	≥70	*p* Value *
**No**	110	216	302	135	
**MoCA mean score**	27.40 ± 2.3	26.29 ± 2.6	25.58 ± 2.4	24.81 ± 2.9	<0.001
**DSST**	74.54 ± 14.4	64.64 ± 13.4	56.63 ± 11.6	49.26 ± 12.5	<0.001
**Fazekas mean score (WMH)**	0.74 ± 0.8	1.22 ± 1.0	1.73 ± 1.1	2.57 ± 1.4	<0.001
**WMH Volume**	1.21 ± 0.8	1.54 ± 1.3	1.94 ± 1.4	3.23 ± 4.6	<0.001
**GM Volume**	622.69 ± 48.7	588.0 ± 49.7	562.9 ± 46.9	529.6 ±38.3	<0.001
**WM Volume**	542.1 ± 61.4	516.6 ± 61.5	493.9 ± 54.3	458.7 ± 43.7	<0.001

Abbreviations: MoCA = Montreal Cognitive Assessment; DSST = Digit Symbol Substitution Test; WMH = white matter hyperintensities; GM = grey matter; WM = white matter. Continuous values are mean (SD). Brain volumes are expressed in mm^3^. * Kruskal–Wallis rank-sum test. In post hoc, there were significant differences between all age groups.

**Table 3 ijerph-19-12228-t003:** Results of mediation analysis.

	Effect	Coefficient	SE	z-Value	*p*-Value
**CR~MoCA (through WMH vol and WM vol)**	Direct effect of CR on MoCA (c)	**0.286**	**0.068**	**4.224**	**<0.001**
	Direct effect of CR on WMH (a_1_)	**−0.142**	**0.064**	**−2.220**	**0.03**
	Direct effect of CR on WM vol (a_2_)	0.029	0.023	1.287	0.198
	Direct effect of WMH on MoCA (b_1_)	**−0.187**	**0.039**	**−4.841**	**<0.001**
	Direct effect of WM on MoCA (b_2_)	**0.420**	**0.091**	**4.628**	**<0.001**
	Indirect effect (through WMH) of CR on MoCA (a_1_b_1_)	**0.026**	**0.013**	**2.018**	**0.04**
	Indirect effect (through WM vol) of CR on MoCA (a_2_b_2_)	0.012	0.010	1.240	0.215
	Total effect of CR on MoCA (a_1_b_1_ + a_2_b_2_ + c)	**0.324**	**0.067**	**4.826**	**<0.001**
**CR~MoCA (through WMH vol and GM vol)**	Direct effect of CR on MoCA (c)	**0.270**	**0.068**	**3.987**	**<0.001**
	Direct effect of CR on WMH (a_1_)	**−0.170**	**0.064**	**−2.631**	**0.009**
	Direct effect of CR on GM vol (a_2_)	**0.054**	**0.021**	**2.567**	**0.01**
	Direct effect of WMH on MoCA (b_1_)	**−0/161**	**0.038**	**−4.207**	**<0.001**
	Direct effect of GM on MoCA (b_2_)	**0.549**	**0.090**	**6.086**	**<0.001**
	Indirect effect (through WMH) of CR on MoCA (a_1_b_1_)	**0.027**	**0.012**	**2.231**	**0.026**
	Indirect effect (through GM vol) of CR on MoCA (a_2_b_2_)	**0.030**	**0.013**	**2.365**	**0.018**
	Total effect of CR on MoCA (a_1_b_1_ + a_2_b_2_ + c)	**0.327**	**0.067**	**4.897**	**<0.001**
**CR~DSST (through WMH vol and WM vol)**	Direct effect of CR on DSST (c)	**1.780**	**0.327**	**5.447**	**<0.001**
	Direct effect of CR on WMH (a_1_)	**−0.142**	**0.064**	**−2.220**	**0.03**
	Direct effect of CR on WM vol (a_2_)	0.029	0.023	1.287	0.198
	Direct effect of WMH on DSST (b_1_)	**−1.241**	**0.212**	**−5.865**	**<0.001**
	Direct effect of WM on DSST (b_2_)	**3.042**	**0.498**	**6.107**	**<0.001**
	Indirect effect (through WMH) of CR on DSST (a_1_b_1_)	**1.176**	**0.085**	**2.076**	**0.04**
	Indirect effect (through WM vol) of CR on DSST (a_2_b_2_)	0.089	0.070	1.260	0.208
	Total effect of CR on DSST (a_1_b_1_ + a_2_b_2_ + c)	**2.044**	**0.317**	**6.454**	**<0.001**
**CR~DSST (through WMH vol and GM vol)**	Direct effect of CR on DSST (c)	**1.649**	**0.328**	**5.032**	**<0.001**
	Direct effect of CR on WMH (a_1_)	**−0.170**	**0.064**	**−2.631**	**0.009**
	Direct effect of CR on GM vol (a_2_)	**0.054**	**0.021**	**2.567**	**0.01**
	Direct effect of WMH on DSST (b_1_)	**−1.052**	**0.208**	**−5.060**	**<0.001**
	Direct effect of GM on DSST (b_2_)	**4.088**	**0.491**	**8.318**	**<0.001**
	Indirect effect (through WMH) of CR on DSST (a_1_b_1_)	**0.179**	**0.076**	**2.334**	**0.02**
	Indirect effect (through GM vol) of CR on DSST (a_2_b_2_)	**0.223**	**0.091**	**2.453**	**0.01**
	Total effect of CR on DSST (a_1_b_1_ + a_2_b_2_ + c)	**2.051**	**0.317**	**6.475**	**<0.001**

Abbreviations: CR = cognitive reserve; MoCA = Montreal Cognitive Assessment; DSST = Digit Symbol Substitution Test; WMH = white matter hyperintensities; GM = grey matter; WM = white matter. a1, b1, a2, b2, c are illustrated at Figure 1. Continuous values are mean (SD). Significant effects were marked with bold characters and adjusted for age, sex, and education.

## Data Availability

Not applicable.

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
