# Peer review of "Unraveling the Protective Effects of Cognitive Reserve on Cognition and Brain: A Cross-Sectional Study"

_ijerph, 2022, doi:10.3390/ijerph191912228_

Round 1
Reviewer 1 Report
This paper seeks to contribute to the conversation about cognitive reserves by looking at the mediating role of brain structures on the relation between cognitive reserve and cognitive performance.
This is a well-written paper (minus a few typos) that speaks to a highly important topic: the sources of the relation between cognitive reserves (intelligence, education, occupation, physical activity, cognitive activity, social engagement, etc.) and cognitive performance (e.g., assessed with the Montreal Cognitive Assessment).
I enjoyed reading the paper and recommend publication. The only improvement i suggest is to look again through the discussion and present some of the cited work already in the introduction, namely to better understand the research in the field before the study is presented. For example, it becomes apparent only in the discussion that the obtained results conflict somewhat with existing research. Better to discuss the relevant existing research already in the introduction (also to avoid having overly long paragraphs in the discussion).
Reviewer 2 Report
Thank you for the opportunity to review the article entitled “Unraveling the protective effects of cognitive reserve on cognition and brain: A cross-sectional Study.” This is a cross-sectional study of 763 participants (mean age = 61 years; 481 women) from the Prospective Urban Rural Epidemiology Study located in Poland. Cognitive reserve was measured using a sum score indicating participation in social groups, being married or in a partnership, and engagement in moderate or vigorous physical activity in leisure time, being employed, graduation from high school or trade school, and graduation from college or university. Cognition was measured using the MoCA and DSST. Measures of brain reserve included total gray matter and total white matter, while measures of brain maintenance included white matter hyperintensities.
Introduction:
Comment 1: I believe you meant to provide a definition on Page 2 for brain reserve.
Comment 2: As a suggestion, there may be a need to have a sentence preparing the reader to talk about definitions for three constructs early on as the paragraph is very dense. For example, “Since dementia research has focused more on preventive and protective factors of cognitive impairments in recent 40 years, cognitive reserve (CR) has become an essential phenomenon under investigation as has relations to brain reserve (BR) and brain maintenance (BM).”
Comment 3: Can you provide citations for this sentence: “However, until now, the central attempts to estimate cognitive reserve on cognition have often been limited to a single marker analysis.”
Methods
Comment 4: In the methods, can you justify your inclusion of the cognitive reserve components by re-citing the literature you use in the introduction covering each component?
Comment 5: Please define how the measures are capturing brain reserve and brain maintenance. I see them later as brain reserve being WM and GM volume and brain maintenance being represented by WMH.
Comment 6: For the statistical analyses can you describe what R packages were used for the mediation model in version 4.1.2? Did the package consider bootstrapping of the indirect effects, such as provided in the Preacher and Hayes mediation software?
Comment 7: I think Figure 1 is the wrong figure included in the manuscript. Seems that the figure should be a study flowchart and not one from a literature review.
Discussion
Comment 8: I believe it should be highlighted that the innovation of the study is the measure of cognitive reserve. Previous measures have been only single items and newer measures have severe problems (i.e., residualization), for example, see Elman et al., 2022. Have any other papers used this approach?
Reference: Elman, J. A., Vogel, J. W., Bocancea, D. I., Ossenkoppele, R., van Loenhoud, A. C., Tu, X. M., & Kremen, W. S. (2022). Issues and recommendations for the residual approach to quantifying cognitive resilience and reserve. Alzheimer's Research & Therapy, 14(1), 1-10.
Comment 9: Also in the discussion please specify how this study differs from previous work as specifically as possible, especially in regard to looking a total gray matter volume and total white matter volume. The authors do a good job explaining the previous work with white matter hyperintensities.
Comment 10: In the discussion, the authors seem to argue that the lack of complete mediation supports cognitive reserve as a unique construct. This does seem the case given the significant association between cognitive reserve and cognition after adjusting for brain reserve and brain maintenance, but I wanted to caution against interpreting a complete mediation as the idea that cognitive reserve does not exist. If these lifestyle behaviors and opportunities afford cognition in later life absent or in the presence of disease, we would still call this cognitive reserve, yes? Instead, I think the authors are highlighting that there is a unique component of cognitive reserve that remains to be described (i.e., neural compensation). This would also be more consistent with the original phrasing of the proposed analyses in the introduction to examine the mechanisms of cognitive reserve on cognition.
Comment 11: In the discussion, I would ask the authors to consider describing the timing of cognitive reserve which may be best for modification. Currently, it may read as if providing education in later life may reduce dementia risk, but surely changes in the brain related to cognitive reserve occur much earlier in life? Any thoughts from the authors on timing and how this relates to cognitive resilience would be helpful.
